# Effect of periodontal treatment on the glomerular filtration rate, reduction of inflammatory markers and mortality in patients with chronic kidney disease: A systematic review

Théo Delbove[1,2,3], François Gueyffier[4], Laurent Juillard[5], Emilie Kalbacher[5], Delphine Maucort-Boulch[6], Patrice Nony[4], Brigitte Grosgogeat[1,2,7], Kerstin Gritsch[1,2,3]*

1 Laboratoire des Multimatériaux et Interfaces UMR CNRS 5615, Université Claude Bernard Lyon 1, Université de Lyon, Villeurbanne, France, 2 UFR d'Odontologie, Université Claude Bernard Lyon 1, Université de Lyon, Lyon, France, 3 Service d'Odontologie (Unité Fonctionnelle de Parodontologie), Hospices Civils de Lyon, Lyon, France, 4 Service de Pharmacotoxicologie–Unité de Pharmacologie Clinique et Essais Thérapeutiques (UPCET), Lyon, France, 5 Service de Néphrologie, Hôpital Edouard Herriot, Hospices Civils de Lyon, /INSERM U1060 CARMEN Research Unit, Lyon, France, 6 Service de Biostatistiques, Hospices Civils de Lyon, Lyon, France, 7 Service d'Odontologie (Unité Fonctionnelle de Recherche Clinique), Hospices Civils de Lyon, Lyon, France

* kerstin.gritsch@univ-lyon1.fr

## Abstract

### Aim

To assess the effect of periodontal treatment (PT) on glomerular filtration rate (GFR), systemic inflammation, or mortality in patients with chronic kidney disease (CKD).

### Methods

A literature search was performed on PubMed and Web of Science databases on articles published until December 2019. The PRISMA guidelines were used throughout the manuscript.

### Results

Of the total studies found, only 18 met the inclusion criteria; four retrospective and 14 prospective studies (including 3 randomized controlled trials–RCT). After PT, 3 studies investigated GFR, 2 found significant improvement; 11 (including 2 RCTs) investigated C-reactive protein levels, 9 found a significant improvement (including the 2 RCTs); 5 (including 3 RCTs) investigated Interleukine-6 level, 4 found a significant improvement (including 2 RCTs) and 2 studies evaluated mortality, one (retrospective study) found a significant difference.

### Conclusions

Within the limitations of the present study, PT seems to improve CKD status, especially by reducing the systemic inflammation. Further RCTs are needed to confirm the results and

**Data Availability Statement:** All relevant data are within the manuscript and its Supporting Information files.

**Funding:** The author(s) received no specific funding for this work.

**Competing interests:** The authors have declared that no competing interests exist.

specifically assess the influence of different types of PT in CKD patients. Taking into consideration the ability of PT to prevent further tooth loss and denutrition, early management of periodontitis is extremely important in patients with impaired renal function.

## Introduction

Periodontitis is an immune-mediated inflammatory disease that is due to bacterial infection (periodontal flora), leading to the destruction of the tissues surrounding the tooth (periodontium). The global prevalence of its severe forms has been estimated to 10.8%, affecting 743 million people worldwide [1]. Both innate and adaptive immune responses are stimulated by the periodontal bacteria [2] and the release of bacterial products and inflammatory mediators from periodontal pockets result in low-grade systemic inflammation [3, 4]. Periodontitis has been most strongly associated with an increase of pro-inflammatory cytokines levels [5] and serum C-reactive protein (CRP) levels [6]. Moreover, the relationship between periodontitis and some systemic diseases has been well documented (for example periodontitis and diabetes [7, 8], periodontitis and cardiovascular diseases [9, 10]). There is also strong evidence supporting a positive association between periodontitis and chronic kidney disease (CKD) [11–13], and some studies suggest a more severe periodontal breakdown with the progression of kidney disease [14, 15], and the duration of dialysis therapy [16]. Furthermore, the risk of death is higher in haemodialysis (end-stage renal disease) patients with periodontitis than those without periodontitis [17, 18]. If relationship between these two diseases is often described, recent meta-analysis concluded that there is a need to determine whether prevention of periodontal disease or periodontal treatment can reduce the severity of CKD [11, 19]. CKD is related to many specifics markers: diagnostic of CKD is based on the glomerular filtration rate (GFR) levels, also CKD have an influence in inflammatory markers, and is related to death due to cardio-vascular disease. We aimed to systematically review all prospective and retrospective studies assessing the effect of periodontal treatment in CKD using clinical parameters, in patients suffering of periodontitis.

## Materials and methods

The preferred reporting items for systematic reviews and meta-analyses (PRISMA) [20] were used as a guideline throughout the manuscript and the protocol was registered in the international prospective register of systematic reviews (PROSPERO; registration number CRD42017075376).

### Study selection

The inclusion criteria were: at least one group of patients with periodontitis and CKD at any stage, including end-stage-renal disease (ESRD); any kind of PT, including oral hygiene instruction, nonsurgical or surgical treatment; and at least one of the following–renal function evaluation (i.e. GFR), inflammatory marker, or mortality.

### Search strategy

A literature search was performed using two electronic databases (PubMed & Web of Science) on articles published until December 2019. Language restrictions were not applied. The keywords used in the electronic search (in "Title/Abstract" for MEDLINE database and "Topic" for Web of Science database) were as follows: periodont* AND (oral hygiene OR root planing

OR treatment OR therapy) AND (end-stage renal disease OR renal failure OR chronic kidney disease OR hemodialysis OR dialysis)

Authors (TD, KG) independently reviewed and screened the title and abstract of potentially relevant records and determined final eligibility through examination of full texts. Citations were screened for additional publications that were missed by the electronic search. Review articles and short communications were excluded. Disagreements were resolved through discussion.

## Data collection

Our analysis focused for each study on the protocol parameters, the assessment criteria, the comparison groups, and the biological and clinical results. To determine the bias of studies, the Cochrane risk of bias tool was used for RCTs [21]; two reviewers (TD, PN) assigned a judgment of high, low, or unclear risk of material bias for each item independently. Discrepancies about ~~risk of~~ each risk of bias have been resolved by discussion in the first instance; where consensus could not be reached, a third reviewer was consulted. The quality of each non-randomized study (including at least one control group) was evaluated using the Newcastle-Ottawa Scale [22]. This scale includes 3 categorical criteria with a maximum score of 9 points: the selection of the study groups; the comparability of the groups; and the ascertainment of the outcome of interest. The quality of each study was rated using the following scoring algorithms: ≥7 points was considered as "high", 4 to 6 points was considered as "moderate", and ≤3 point was considered as "poor" quality study.

## Results

### Study selection

The search identified 369 records. After removal of duplicates 277 records remained and the title and abstract of these records were assessed. A total of 257 records were rejected because they did not meet the inclusion criteria. The remaining 20 underwent full text examination: one article did not meet the inclusion criteria (publication of a future study protocol) [23] and one article was a short communication [24]; 18 studies were finally included (Fig 1) [25–42].

### Study characteristics

**Type of studies.**   Among the 18 included studies, 4 were retrospective [39–42], and 14 were prospective studies [25–38] including 3 RCTs (Table 1) [27, 29, 34].

**Study design.**   There were 17 studies that had an experimental group mixing renal disease, periodontitis and periodontal treatment [25–41], and one had a treatment cohort combining periodontitis and periodontal treatment and assessed the incidence and risk of ESRD in this group [42].

In prospective studies, seven did not include control groups [25, 28, 31–33, 35, 38], while the remaining 7 had at least one control group [26, 27, 29, 30, 34, 36, 37] with 3 studies including two control groups [26, 27, 30]. In these 7 studies including control groups, the experimental groups consisted of patients with renal disease (different stages), periodontitis and periodontal treatment [27, 29, 30, 34, 36, 37] or of patients with diabetes, dialysis, periodontitis and periodontal treatment [26]. In those studies, the control groups consisted of patients with periodontitis and periodontal treatment without renal disease in 4 studies [26, 27, 36, 37], patients with renal disease and periodontitis in 3 studies [29, 30, 34], patients with renal disease, periodontitis and periodontal treatment limited to oral hygiene instructions in 1 study [27], patients with diabetes, periodontal disease and periodontal treatment in one study [26] or patients with renal disease in the last study [30].

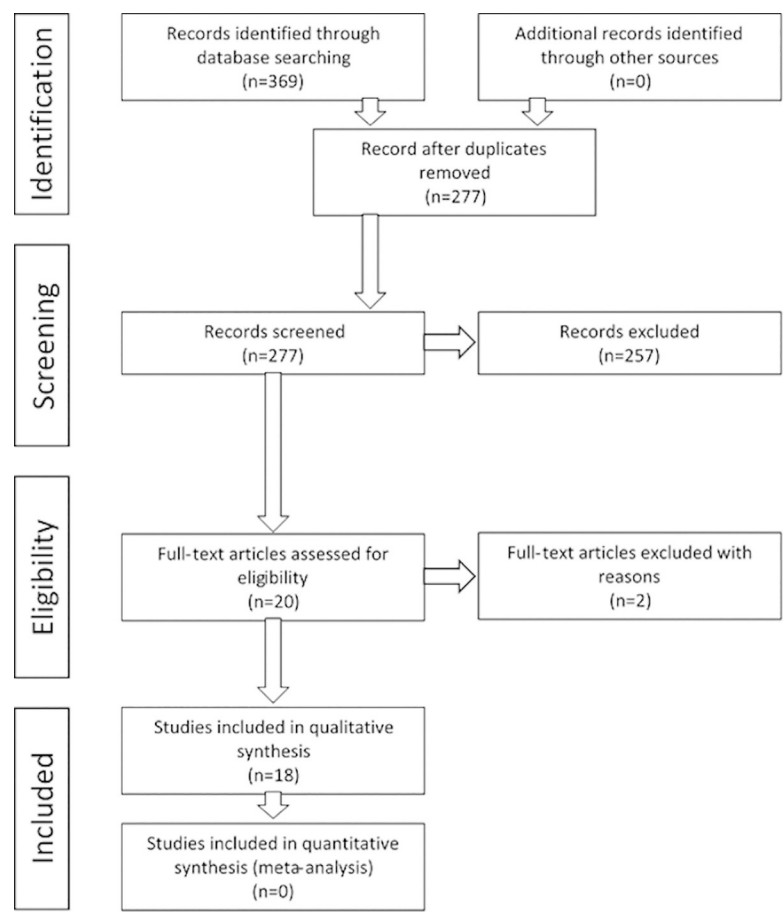

**Fig 1. Preferred Reporting Items for Systematic Reviews and Meta-Analyses (PRISMA) flow diagram.**

In the retrospective studies, three estimated the risk of complications of patients with end-stage renal disease, periodontitis and periodontal treatment (experimental group) versus end-stage renal disease (control group) [39–41] and the other estimated the incidence and risk of ESRD in patients with periodontitis and periodontal treatment (experimental group) versus patients with periodontal disease only (without periodontal treatment) (Table 1) [42].

**Primary endpoint.** The primary endpoint of 9 studies was systemic inflammation: eight studies evaluated this using CRP or high-sensitivity CRP (hs-CRP) levels [27, 29, 31–33, 35, 36, 38] and 1 study evaluated this using interleukin-6 [34]. Five studies [30, 39–42] focused on clinical outcomes; ESRD incidence [42], the risk of hospitalization for an infectious disease [41] or cardiovascular diseases [39, 40], and mortality [30, 39, 40]. For 2 studies [28, 37], the main assessment criterion was the GFR estimated from the Modification of Diet in Renal Disease (MDRD) [28] or the Cockcroft and Gault equation [37]. Two studies did not define primary endpoint (Table 2) [25, 26].

**Assessment of renal disease and systemic inflammation.** To evaluate the impact of periodontal treatment on CKD, the included prospective studies evaluated 22 different criteria related to renal dysfunction and/or systemic inflammation (Table 2); the most frequently evaluated was CRP (11 studies), followed by albumin (10 studies), cholesterol (high/low-density lipoprotein, HDL/LDL) (7 studies), transferrin (6 studies), haemoglobin (6 studies), phosphate (6 studies), calcium (6 studies), parathyroid hormone (5 studies), ferritin (5 studies), IL-6 (5

**Table 1. Characteristics of the 18 included studies.**

| Author, year | Experimental group (number of patients) | Control group (number of patients) | Periodontal Treatment | Assessed criteria | Main results | Study design and other methological aspects |
|---|---|---|---|---|---|---|
| Rapone et al. [25] | ESRD[a] + PD + PT (n = 66) | - | Non surgical periodontal treatment (mechanical supra- and sub-gingival debridement and root planing by quadrant using hand and ultrasonic instruments under local anaesthesia. performed at baseline and at 3 months. Follow-up at 3 and 6 months. | Serum albumin (mg/dL) Serum creatinine (mg/dL) CRP PI (Löe) BoP PD (mm) CAL (mm) | At 3 months after PT All periodontal parameters were improved vs baseline (p<0.001). There was significant correlation between PI and CRP (p = 0.012), BoP and CRP (p = 0.002) and CAL and CRP (p = 0.022). There was no significant correlation between periodontal parameters and albumin and creatinine levels. At 6 months after PT Serum albumin decreased vs baseline (p<0.001). Serum creatinine increased vs baseline (p = 0.002). All periodontal parameters were improved vs baseline (p<0.001). There was significant correlation between PI and CRP (p<0.001), BoP and CRP (p<0.001). There was no significant correlation between periodontal parameters and albumin and creatinine levels. | Cohort study No control group Single center study Type I error rate inflation Exploratory results |
| Tasdemir et al. [26] | CKD[b] + Diabetes + PD + PT (n = 20) | (1) PD + PT (n = 20) (2) Diabetes + PD + PT (n = 20) | All groups received standard oral hygiene instruction and full-mouth scaling and root planing (protocol performed based on Quirynen et al. 1995). Oral hygiene control and reinstructions were provided during all visits. Scaling and root planing was repeated if necessary at the end of the study. During the PT sessions, necessary tooth extractions were performed and referrals for endodontic and restorative treatments were given immediately after PT was completed. | TNF-α (pg/mL) IL-6 (pg/mL) PTX-3 (ng/mL) Hs-CRP Ferritin TC HDL LDL HbA1c Glucose Insulin Homa-Ir Alb Cr BUN PTH Ca Phosphorus serum Uric acid Hb PI (Silness and Loë) GI (Loë and Silness) BOP (%) PD (mm) CAL (mm) DMFT (index) Decay (n) Missing teeth(n) | Experimental group: TNF-alpha, PTX-3, IL-6 and Hs-CRP values significantly decreased at 3 months after PT compared to baseline (p<0.001). Glucose levels and Homa-Ir significantly decreased at 3 months after PT compared to baseline (p = 0.042 and p = 0.037, respectively). All periodontal parameters were improved at 3 months after PT compared to baseline (p<0.001) No significant changes were noticed for the other parameters compared to baseline. Control group (1): TNF-alpha values significantly decreased at 3 months after PT compared to baseline (p = 0.022). All periodontal parameters were improved at 3 months after PT vs baseline (p<0.001). No significant changes were noticed for the other parameters compared to baseline. Control group (2): TNF-alpha values significantly decreased at 3 months after PT compared to baseline (p = 0.037). All periodontal parameters were improved at 3 months after PT vs baseline (p<0.001) No significant changes were noticed for the other parameters compared to baseline. All the periodontal parameters were similar at baseline in all groups. There were no significant differences among groups after at 3 months after PT. | **Non randomized 3 groups study** Before/after clinical study, Single center study Significant differences in terms of gender among the groups Type I error rate inflation Exploratory results |
| Guo and Lin et al. [27] | CKD[c] + PD + PT (n = 26) | (1) PD + PT (n = 30) (2) CKD[a] + PD + PT (n = 27) | Experimental group and control group (1): oral hygiene instruction and nonsurgical periodontal treatment (ultrasonic scaling, subgingival scaling and root planing) Control group (2): oral hygiene instruction | hs-CRP (mg/L) IL-6 (pg/ml) TNF-α (pg/ml) PPD (mm) BOP (%) CAL (mm) | Experimental group: for all assessed criteria, the values were significantly lower at 6 weeks after PT compared to baseline (p< 0.05). Control group (1): for all assessed criteria, the values were significantly lower at 6 weeks after PT compared to baseline (p< 0.05). Control group (2): hs-CRP, BOP and CAL values were significantly lower at 6 weeks after PT compared to baseline (p< 0.05). No significant changes were noticed for IL-6, TNF-α and PPD values before and after PT. | **RCT** Single center study Type I error rate inflation Exploratory results |
| Almeida et al. [28] | CKD[d] + PD + PT (n = 26) | - | Oral hygiene instruction and nonsurgical periodontal treatment (manual and sonic supragingival scaling, subgingival scaling and root planing) in four to six sessions for two weeks. Oral hygiene instruction reinforced at each session. | eGFR (mL/min/1.73m²) albumin(g/dL) cholesterol (mg/dL) TG (mg/dL) ADMA (μmol/mL) PI (%) PPD (4-5mm) (% sites) PPD (≥ 6mm) (% sites) BOP (%) CAL (4-5mm) (% sites) CAL (≥ 6mm) (% sites) | At 3 months after PT eGFR values significantly increased compared to baseline (p<0.01). PI, PPD (4–5 mm), PPD (≥ 6mm), BOP, CAL (4-5mm) and CAL (≥ 6mm) values significantly decreased compared to baseline (p< 0.001). No significant changes were noticed for albumin, cholesterol and ADMA values before and after PT. At 6 months after PT: eGFR values significantly increased compared to baseline (p<0.05). ADMA values significantly decreased compared to baseline (p<0.05). PI, PPD (4–5 mm), PPD (≥ 6mm), BOP, CAL (4-5mm) and CAL (≥ 6mm) values significantly decreased compared to baseline (p< 0.001). No significant changes were noticed for albumin and cholesterol values before and after PT. | **Cohort study** No control group Single center study Type I error rate inflation Exploratory results |

*(Continued)*

**Table 1.** (Continued)

| Author, year | Experimental group (number of patients) | Control group (number of patients) | Periodontal Treatment | Assessed criteria | Main results | Study design and other methological aspects |
|---|---|---|---|---|---|---|
| Fang et al. [29] | ESRD + PD + PT (n = 48) | ESRD + PD (n = 49) | Experimental group: oral hygiene instruction, supra- and sub-gingival instrumentation (scaling and root planing) using ultrasonic and manual instruments completed within 24 hours. Teeth extraction if necessary. Supragingival prophylaxis at 3 months. Control group: no therapy measures or formal oral hygiene instructions | hs-CRP IL-6 (pg/ml) albumin (g/dl) HDL-C (mmol/l) LDL-C (mmol/l) Tf (g/l) TG (mmol/l) BUN (mg/dl) nPCR (g/kg/day) Cr (mg/dl) TNF-α (pg/ml) ALC (103/μl) PI (Silness and Loë) PPD (mm) BOP (%) CAL (mm) | At 6 weeks after PT: hs-CRP values decreased compared to baseline and were lower to those of the control group without significance (p = 0.196). Compared to baseline values, albumin, Tf and BUN values significantly increased (p<0.05) whereas TNF-α values decreased (p<0.05). For HDL-C and Cr values, results showed an increase compared to both baseline and control group values (p<0.05). No significant changes were noticed for IL-6, LDL-C, TG, nPCR and ALC values before and after PT or compared to the control group. All periodontal parameters values significantly decreased compared to baseline and to the control group (p<0.05). At 3 months after PT: hs-CRP values decreased compared to baseline and were significantly lower to those of the control group (p < 0.01). Compared to both baseline and control group values, significant differences (p<0.05) were noticed for IL-6 values (decrease) and for albumin, Tf, BUN and Cr values (increase). No significant changes were noticed for HDL-C, LDL-C, TG, nPCR, TNF- α and ALC values before and after PT or compared to the control group. All periodontal criteria values significantly decreased compared to baseline (p<0.05). At 6 months after PT: Same results than after 3 months with values even lower for hs-CRP compared to the control group (p<0.001) and all periodontal criteria significantly decreased compared to baseline and control group values (p < 0.05). | **RCT** Single center study For primary endpoint, no statistical analysis specific to repeated measurements Type I error rate inflation for secondary endpoints |
| De Souza et al. [30] | ESRD + PD + PT (n = 43) | (1) ESRD+ PD (n = 30) (2) ESRD (n = 49) | Experimental group: Oral hygiene instruction, antibiotic prophylaxis (2g amoxicillin), and scaling and root planing using hand and ultrasonic instrumentation. Variable number of follow-up visits with additional treatment. Control group (1): patients that were invited to receive treatment but did no show up for the treatment visits | Fatal events | 34 fatal events during the follow-up time (mean 64.1 ± 11.2 months): 15 for experimental group, 11 for control group (1) and 8 for control group (2). No significant differences between the groups, after adjustments for confounders in the multivariate model. | **Prospective operational study** Observational study Single center study Exploratory results |
| Kocyigit et al. [31] | ESRD+ PD[e] + PT (n = 43) | - | Oral hygiene instructions (alone for healthy patients), full-mouth scaling and root planing with periodontal curettes and ultrasonic intrumentation in 2 or 4 sessions. Patients instructed to rinse twice daily for 1 min with 0.12% solution of chlorhexidine digluconate for 14 days. During PT, necessary tooth extractions were performed. | hs-CRP (mg/L) White blood cell count Hemoglobin (g/dL) Fasting glucose (mg/dL) Total cholesterol (mg/dL) Triglyceride (mg/dL) LDL cholesterol (mg/dL) HDL cholesterol (mg/dL) Serum calcium (mg/dL) Serum phosphorus (mg/dL) Serum albumin (g/dL) Transferrin saturation (%) hs-CRP (mg/L) Intact parathormone (pg/mL) Kt/V$_{urea}$ Plaque index Gingival index Pocket depth Bleeding on probing (%) Clinical attachment loss Gingival recession | At 3 months after PT _Healthy/gingivitis_: No significant changes were noticed for all parameters _Slight-to-moderate periodontitis_: hs-CRP values decreased compared to baseline CAL and PI decrease compare to baseline (p = 0.014) No significant changes were noticed for the other parameters. _Severe periodontitis_: hs-CRP values decreased compare to baseline (p = 0.001) CAL and PI decrease compare to baseline (p = 0.001) No significant changes were noticed for the other parameters. Comparison according to severity of PD: Significant difference between healthy/gingivitis and slightly moderate periodontitis groups for White blood cell count (p = 0.031). Hs-CRP, GI, PPD, CAL, Gingival recession, showed significant difference between healthy/gingivitis and slightly moderate (p = 0.023) and severe periodontitis groups (p<0.001). | **Cohort study** Observational study Single center study Exploratory results |

_(Continued)_

**Table 1.** (Continued)

| Author, year | Experimental group (number of patients) | Control group (number of patients) | Periodontal Treatment | Assessed criteria | Main results | Study design and other methological aspects |
|---|---|---|---|---|---|---|
| Siribamrungwong et al. [32] | ESRD[f] + PD + PT (n = 15) | - | Oral hygiene instruction and nonsurgical periodontal treatment (scaling, root planning and curettage in patients presenting teeth with periodontitis). Surgical treatment consisting of gingivectomy or gingivoplasty 6 months after the non-surgical phase in patients presenting teeth with attachment loss ≥ 3 mm | hs-CRP (mg/L) Ferritin (ng/mL) Blood urea nitrogen (mg/dL) Albumin (g/dL) Total cholesterol (mg/dL) Hemoglobin (g/dL) Calcium (mg/dL) Phosphate (mg/dL) Parathyroid hormone (pg/mL) Transferrin saturation (%) $Kt/V_{urea}$ nPCR Erythropoietin dosage (unit/week) PI (plaque index) PDI (periodontal disease index) | At 16 weeks after PT: Compared to baseline values, hs-CRP and erythropoietin values significantly decreased (p<0.05) whereas blood urea nitrogen values increased (p<0.05). Periodontal parameters were significantly improved. | **Interventional uncontrolled study** Observational study Single center study Exploratory results |
| Yazdi et al. [33] | ESRD + PD[g] + PT (n = 77) | - | According to the need of each patient: oral hygiene instruction, scaling and root planing with ultrasound devices and manual instruments, polishing, and subgingival curettage in 1h sessions repeated according to the need of each patient. Teeth extraction and restoration of carious lesions if needed. A follow-up program was carried out based on the need of the patients (additional ora hygiene instructions and supragingival prophylaxis) | hs-CRP (mg/L) | At 8 weeks after initial PT : CRP values showed significant reduction compared to baseline (p< 0.001). No significant relation was found between this difference and the severity of periodontitis. | **Interventional uncontrolled study** No control group Single center study Exploratory results |
| Wehmeyer et al. [34] | ESRD + PD + PT (n = 25) | ESRD + PD (n = 26) | Scaling and root planig under local anesthesia using hand and ultrasonic instruments. Adjunctive local-delivery antimicrobial therapy with controlled-release microsphere-encapsulated biodegradable minocycline to all sites > 5-mm probing depths (at the time of instrumentation and at the 3- and 6-months follow-up appointments). Teeth extraction at the time of scaling and root planning. Treatment done in 1 or 2 appointments. All participants received verbal and written oral hygiene instruction at each visit (instructions and demonstrations). Experimental group: follow-up examinations at 3 and 6 months after completion of treatment Control group: follow-up examinations at 3 and 6 months after baseline (received treatment at the conclusion of the study) | Serum albumin hs-IL-6 (pg/mL) Probing depth Clinical attachement level Bleeding on probin Gingival index Plaque index | At 3 months after PT: No statistical differences for serum albumin or hs-IL-6 level between the groups. About periodontal criteria, statistically significant improvement for the treatment group compared to the control group for the following periodontal parameters: probing depth (p> 0.05), extent of probing depth ≥ 4 mm (p < 0.05), extent of gingival index ≥ 1 (p = 0.01). At 6 months after PT: No statistical differences for serum albumin or hs-IL-6 level between the groups. About periodontal criteria, statistically significant improvement for the treatment group compared to the control group for extent of probing depth ≥ 4 mm only (p < 0.05). | **RCT** Pilot RCT Patients poorly representative Type I error rate inflation Exploratory results |

(*Continued*)

**Table 1.** (Continued)

| Author, year | Experimental group (number of patients) | Control group (number of patients) | Periodontal Treatment | Assessed criteria | Main results | Study design and other methological aspects |
|---|---|---|---|---|---|---|
| Siribamrungwong et al. [35] | ESRD[h] + PD + PT (n = 30 ; 63% of presenting periodontitis) | - | Same as Siribamrungwong et al. [31] | hs-CRP (mg/L) Ferritin (ng/mL) Blood urea nitrogen (mg/dL) Albumin (g/dL) Total cholesterol (mg/dL) Hemoglobin (g/dL) Hematocrit (%) Total leukocyte count ($\mu L^{-1}$) Calcium (mg/dL) Phosphate (mg/dL) Parathyroid hormone (pg/mL) Transferrin saturation (%) $Kt/V_{urea}$ nPCR Erythropoietin dosage (unit/week) PI (plaque index) PDI (periodontal disease index) | After completion of periodontal therapy (duration 6 ± 2 weeks): Significant decrease of hs-CRP (p < 0.001) and erythropoeitin (p < 0.01), and significant increase of blood urea nitrogen (p < 0.01) and albumin (p < 0.001). No statistical differences (p < 0.05) observed in the other parameters. Significantly decrease of periodontal indices; PI (p = 0.001), PDI (p = 0.001). | **Interventional uncontrolled study** No control group Single center study Exploratory results |
| Vilela et al. [36] | CKD[i] + PD + PT (n = 36) | PD + PT (n = 20) | Both groups received oral hygiene instructions. Radicular scraping and subgingival curettage were performed with manual and ultrasound devices in 1-hour sessions over an average period of 4 weeks for non surgical treatment. Follow-up after 15, 30, 60 and 90 days: instructions on oral hygiene and supragingival prophylaxis. | Prohepcidin (ng/mL) IL-6 (pg/mL) us-CRP (mg/L) Serum Fe ($\mu$g/dL) TSat (%) Ferritin (ng/mL) Hemoglobin (g/dL) BOP (%) PPD (mm) CAL (mm) GI PI Sites with PPD ≥ 5 mm (n) | At baseline: PD was more severe in patients in the experimental group. At 3 months after completion of PT: In both groups, significantly decreases were observed for the following parameters ; prohepcidin (p < 0.05 and p < 0.01 for experimental and control groups, respectively), IL-6 (p < 0.01 and p < 0.05 for experimental and control groups, respectively) and us-CRP (p < 0.01 and p < 0.05 for experimental and control groups, respectively). In the control group, a significant increase in the levels of hemoglobin (p < 0.05) and ferritin (p < 0.05) was also observed. In both groups, significant improvement in periodontal parameters were noticed (p < 0.05) ; BOP, PPD, CAL, GI, PI, sites with PPD ≥ 5 mm. | **Interventional controlled non-randomized** Non-randomized trial Single center study Exploratory results |
| Artese et al. [37] | CKD[j] + PD + PT (n = 21) | PD + PT (n = 19) | Both groups received non-surgical treatment consisting of oral hygiene instructions, scaling and root planning (manual instrumentation), under local anesthesia, by sextant for each dental visit completed over a 6–8 week period for the entire mouth. | GFR (Cockcroft and Gault equation, multiplied by 0.85 for female patients) Serum creatinine GB (Gingival bleeding index, % of sites) VP (Visible plaque index, % of sites) SUP (suppuration, % of sites) BOP (bleeding on probing, % of sites) PD (Probing depth 1-3mm, 4-6mm, ≥ 7mm, % of sites) AL (attachment loss 1-3mm, 4-6mm, ≥ 7mm, % of sites) | At 3 months after PT: Statistically significant improvement of GFR was observed in the experimental group (p < 0.05) and in the control group (p < 0.01). No significant differences were noticed in both groups for Creatinine levels. Both groups experienced a significant improvement in many of the periodontal parameters measured, without significant differences between the groups. | **Interventional controlled non-randomized** Non-randomized trial Single center study Exploratory results |
| Kadiroglu et al. [38] | ESRD[k] + PD + PT + high CRP level (n = 21, 48% gingivitis/52% peiodontitis) | - | All patients received standard oral hygiene instructions before the first session of scaling and root planning (non-surgical therapy). Sessions performed once a week for a month (standard periodontal curettes and an ultrasonic device starting 1 day after the HD session). | CRP (mg/l) ESR (Erythrocyte sedimentation rate, mm/h) Ferritin (ng/mL) Leukocyte (K/UL) rHuEPO (U/week) Hemoglobin (g/dL) Hematocrit (%) | At 4–6 weeks after PT: Significant decrease of CRP level and erythrocyte sedimentation rate (p = 0.001) and significant increase of hemoglobin level and hematocrit level (p < 0.01). | **Interventional** Non-randomized study No control group Single center study Exploratory results No results about periodontal parameters after PT |

(Continued)

**Table 1.** (Continued)

| Author, year | Experimental group (number of patients) | Control group (number of patients) | Periodontal Treatment | Assessed criteria | Main results | Study design and other methological aspects |
|---|---|---|---|---|---|---|
| Santos-Paul et al. [39] | ESRD + PD + PT (n = 206) | ESRD (n = 203) | Debridement of the affected root surfaces or tooth extraction and any type of treatment considered suitable (plaque control, caries, endodontic intervention). Prophylactic antibiotics were not used. All patients were given instructions in basic oral hygiene. Patients were then referred to generalists for continuing dental health assessment and were followed-up for 24 months or until death or renal transplantation. | Incidence of cardiovascular events and death | Periodontal treatment was associated with reduction in cardiovascular events (HR 0.43; 95% CI 0.22–0.87), coronary events (HR 0.31; 95% CI 0.12–0.83), and cardiovascular deaths (HR 0.43; 95% CI 0.19–0.98). | **Single center observational cohort study.** Comparison with historical controls. Periodontal status not assessed in the historical control group Patients in the experimental group were younger. More patients received beta-blockers or renin-angiotensin system inhibitors in the control group. Type I error rate inflation Exploratory results |
| Huang et al. [40] | ESRD + PD + PT (n = 3613) | ESRD (n = 3613) | Subgingival curettage, (scaling and root planing) and periodontal flap surgery. | Incidence of cardiovascular diseases and mortality | Compared with the comparison cohort, the adjusted hazard ratio of hospitalization for cardiovascular diseases was 0.78 (95% confidence interval = 0.73–0.84, p<0.001) in the treatment cohort. The treatment cohort exhibited significantly lower cumulative incidences of cardiovascular diseases and mortality. | **Retrospective observational cohort study** All useful information not available in the NHIRD, low number of PD patients who received no PT. Periodontal status unclear in the control group. Type I error rate inflation Exploratory results |
| Huang et al. [41] | ESRD + PD + PT (n = 4451) | ESRD (n = 4451) | Subgingival curettage, (scaling and root planing) and periodontal flap surgery. | Hospitalization for infectious disease | The intensive PD treated cohort had a significantly lower risk of acute and subacute infective endocarditis (HR = 0.54, 95% CI = 0.35–0.84, p<0.01), pneumonia (HR = 0.71, 95% CI = 0.65–0.78, p<0.001), and osteomyelitis (HR = 0.77, 95% CI = 0.62–0.96, p<0.05) | **Retrospective study** Periodontal status unclear in the control group |
| Lee et al. [42] | PD + PT (n = 35496) Treatment cohort | PD (n = 141824) No-treatment cohort | Subgingival curettage and/or periodontal flap surgery | Incidence and risk of ESRD | The overall incidence of ESRD was 37% lower in the treatment cohort than in the no-treatment cohort (4.66 vs 7.38 per 10000 person-years), with an adjusted HR of 0.59 (95% CI = 0.46 to 0.75). | **Retrospective study** |

**CKD**: Chronic Kidney Disease, **ESRD**: end-stade renal disease, **PD**: Periodontal disease, **PT**: Periodontal treatment, **RCT**: Randomized clinical trial.

[a] receiving hemodialysis for at least one year

[b] continuous ambulatory peritoneal dialysis with diabetic nephropathy

[c] including ESRD

[d] CKD stages 3 and 4

[e] participants classified as healthy/gingivitis, slight-to-moderate and severe periodontitis

[f] stable peritoneal dialysis

[g] participants classified as gingivitis, mild, moderate and severe periodontitis

[h] stable hemodialysis

[i] CKD patients at stages 3 to 5 undergoing conservative treatment

[j] CKD patients at stages 2 to 4 undergoing conservative treatment

[k] arterioveinous fistula, patients on rHuEPO therapy.

studies), triglycerides (4 studies), blood urea nitrogen (4 studies), creatinine (4 studies), eGFR (3 studies), normalized protein catabolism (3 studies), tumour necrosis factor alpha (3 studies), pentraxin-3 (1 study), haematocrit (1 study), absolute lymphocyte counts level (1 study), asymmetric dimethylarginine (1 study), and mortality (1 study). The number of criteria assessed per study ranged from 1 [33] to 15 [26] (Table 2).

**Table 2. Assessment criteria and improvement after PT compared to baseline in prospective studies.**

| Criteria / Authors | Rapone et al. [25]a | Tasdemir et al. [26]a | Guo and Lin [27]b | Almeida et al. [28] | Fang et al. [29]b | De Souza et al. [30] | Kocyigit et al. [31] | Siribamrungwong et al. [32] | Yazdi et al. [33] | Wehmeyer et al. [34]b | Siribamrungwong et al. [35] | Vilela et al. [36] | Artese et al. [37] | Kadiroglu et al. [38] | Number of studies |
|---|---|---|---|---|---|---|---|---|---|---|---|---|---|---|---|
| Mortality | | | | | | ns60m | | | | | | | | | 1 |
| hs-CRP | ns3,6m | ***3m | *6w | | ns6w **3m ***6m | NC | *3m | *16w | ***8w | | ***6w | *3m | | ***6w | 11 |
| IL-6 | | ***3m | *6w | | ns6w *3,6m | | | | | ns3,6m | | **3m | | | 5 |
| eGFR | | | | **3m *6m | | | | | | | | NC | **3m | | 3 |
| Albumin | ***6m | ns3m | | ns3,6m | *6w,3,6m | NC | ns3m | ns16w | | ns3,6m | ***6w | | | NC | 10 |
| HDL c | | ns3m | | ns3,6m | *6w ns3,6m | | ns3m | ns16w | | | ns6w | | | ns6w | 7c |
| LDL c | | ns3m | | | ns6w, 3,6m | | ns3m | ns16w | | | | | | | 6 |
| Tf | | | | | *6w,3,6m | NC | ns3m | ns16w | | | ns6w | *3m | | NC | 6 |
| Phos | | ns3m | | | | NC | ns3m | ns16w | | | ns6w | | | NC | 6 |
| Hg | | ns3m | | | | NC | | ns16w | | | ns6w | *3m | | **6w | 6 |
| Cal | | ns3m | | | | NC | ns3m | ns16w | | | ns6w | | | NC | 6 |
| PTH | | ns3m | | | | NC | ns3m | ns16w | | | ns6w | | | | 5 |
| Fer | | ns3m | | | | NC | | ns16w | | | ns6w | | | | 5 |
| TC | | ns3m | | ns6m | ns6w, 3,6m | | ns3m | ns16w | | | ns6w | *3m | | NC | 5 |
| BUN | | ns3m | | | *6w, 3,6m | | | *16w | | | **6w | | | | 4 |
| nPCR | | | | | Ns6w,3,6m | | | ns16w | | | ns6w | | | | 4 |
| Cr | **6m | ***3m | | | ***6w, 3,6m | | | | | | | | ns3m | | 3 |
| TNF-a | | ***3m | *6w | | *6w ns3,6m | | | | | | | | | | 3 |
| PTX-3 | | ***3m | | | | | | | | | | | | | 1 |
| ADMA | | | | ns3m *6m | | | | | | | | | | | 1 |
| ALC | | | | | ns6w, 3,6m | | | | | | | | | | 1 |
| Hct | | | | | | | | | | | | | | **6w | 1 |

Primary endpoint always first on column for each study for clear reading, except for Rapone et al. and Tasdemir et al; p value for improvement after periodontal treatment compared to baseline: ns non significant

*p<0.05

**p<0.01

***p<0.001, NC not communicated after periodontal treatment, w week m month

hs-CRP: high sensivity C reactive protein, eGFR: estimated glomerular filtration rate, IL-6: interleukin 6, HDL: high density lipoprotein, LDL: low density Lipoprotein, Hg: Hemoglobin, Phos: phosphate, Tf: transferrin, PTH: parathyroid hormone, Cal: calcium, Fer: ferritin, TC: triglycerid, BUN: blood urea nitrogen, nPCR: normalized protein catabolism, Cr: creatinine, TNF-a: tumor necrosis factor alpha Hct: hematocrit, PTX-3: pentraxin-3, ADMA: asymmetric dimethylarginine, ALC: absolute lymphocyte counts

a Do not defined primary endpoint

b RCT studies

c Cholesterol or HDL & LDL

**Periodontal assessment criteria and periodontal treatment efficacy evaluation.** For 9 of the 14 prospective studies [27–31, 33, 34, 36, 37], diagnosis of periodontitis was conducted by consensus with general chronic periodontitis defined as a CAL >4mm on 30% of affected sites [43]. Three studies [32, 35, 38], including two from the same team, used another system for diagnosis that was based on periodontal disease index (PDI) [44]. For one study [26], the clinical diagnosis of chronic periodontitis was based on the following criteria [45]; the presence of ≥ 5 teeth with ≥ 1 sites with probing depth ≥ 5 mm, clinical attachment level ≥ 2 mm, and the presence of bleeding on probing. For the last study [25], the diagnosis of periodontitis was performed according to the criteria of the American Academy of Periodontology (2017). Among the 14 prospective studies, 11 also evaluated the efficacy of the purposed periodontal treatment, measuring plaque index (PI) [25, 26, 28, 29, 31, 32, 34–37], gingival index (GI) [26, 31, 34, 36, 37], probing pocket depth (PPD) [25–29, 31, 34, 36, 37], bleeding on probing (BOP) [25–29, 31, 34, 36, 37], clinical attachment level (CAL) [25–29, 31, 34, 36, 37], or periodontal disease index (PDI) [32, 35]. Among these 11 studies, 4 showed a significant improvement of 2 criteria or less [31, 32, 34, 35], and 7 of 3 or more criteria (Table 3) [25–29, 36, 37].

For the 4 retrospectives studies [39–42], the authors included patients who underwent for periodontal treatment. No evaluation of the periodontal treatment effectiveness was mentioned.

## Clinical outcomes

Among the 11 studies evaluating CRP levels, 9 showed a significant improvement after periodontal treatment either with respect to baseline [26, 27, 29, 31–33, 35, 36, 38] or to a control group [29]. For IL-6 level, 4 studies (on 5 studies) showed a significant decrease after periodontal treatment compared to baseline [26, 27, 29, 36]. RCT studies comparing experimental groups (ESRD and periodontitis and PT) to control groups (ESRD and PT) were Fang et al. [29] and Wehmeyer et al. [34]. At 3 and 6 months after completion of periodontal treatment, a significant decrease of IL-6 level was demonstrated in the first study [29] whereas no significant differences were observed in the second one (Table 1) [34].

The two studies investigating GFR as primary endpoint found a significant increase after periodontal treatment as compared to baseline [28, 37] without significant differences between the experimental group (CKD and periodontitis and PT) and the control group (PD and PT) (Table 1) [37].

Among the 4 retrospective studies, one found that the incidence of ESRD was 37% lower in the treatment group (PD and PT) than in the non-treatment group (PD without PT) with an adjusted HR of 0.59 (95% CI = 0.46 to 0.75) [42], and the three other found that patients with ESRD and periodontitis receiving periodontal treatment had a significantly lower risk of acute and subacute infective endocarditis, pneumonia, and osteomyelitis [41], and of cardiovascular diseases and mortality [39, 40] than patients with ESRD only (Table 1).

In the prospective study investigating the effect of treatment on fatal events, no significant difference was found between the groups after adjustments for confounders (Table 1) [30].

## Risk of bias

Among the 14 prospective studies, 3 were RCTs [27, 29, 34], 4 included at least one control group [26, 30, 36, 37] and 7 had no control group [25, 28, 31–33, 35, 38].

The 3 RCTs could be used for the Cochrane collaboration's tool for assessing risk of bias in randomized trials. Two studies showed high risk of bias concerned blinding conditions (Table 4) [27, 34].

Table 3. Periodontal criteria and periodontal treatment evaluation after PT in experimental group compared to baseline in prospective studies.

| Criteria / Authors | Rapone et al. [25] | Tasdemir et al. [26] | Guo and Lin [27][a] | Almeida et al. [28] | Fang et al. [29][a] | De Souza et al. [30] | Kocyigit et al. [31] | Siribamrungwong et al. [32] | Yazdi et al. [33] | Wehmeyer et al. [34][a] | Siribamrungwong et al. [35] | Vilela et al. [36] | Artese et al. [37] | Kadiroglu et al. [38] | Number of studies |
|---|---|---|---|---|---|---|---|---|---|---|---|---|---|---|---|
| PI | ***3,6m | ***3m | | ***3,6m | *6w, 3,6m | NC | **3m | *16w | NC | ns3,6m | ***6w | *3m | ***3m | NC | 10 |
| GI | | ***3m | | | | NC | ns3m | | | **3m / -6m | | *3m | *3m | NC | 5 |
| PPD | ***3,6m | ***3m | *6w | ***3,6m | *6w, 3,6m | NC | ns3m | | | *3,6m | | *3m | ***3m | NC | 9 |
| BOP | ***3,6m | ***3m | *6w | ***3,6m | *6w, 3,6m | | ns3m | | NC | ns3,6m | | *3m | ***3m | | 9 |
| CAL | ***3,6m | ***3m | *6w | ***3,6m | *6w, 3,6m | NC | **3m | | NC | ns3,6m | | *3m | ***3m | | 9 |
| PDI | | | | | | | | **16w | | | ***6w | | | NC | 2 |

[a] RCT studies; p value for improvement after periodontal treatment compared to baseline;—non significant

*p<0.05

**p<0.01

***p<0.001, NC not communicated after periodontal treatment, w: weeks, m: months

PI: plaque index, GI: gingival index, PPD: probing pocket depth, BOP: bleeding on probing, CAL: clinical attachment level, PDI: periodontal dental index

**Table 4. Cochrane risk of bias tool's table (RCT studies).**

| Authors | Random sequence generation | Allocation concealment | Blinding of participants and personnel | Blinding of outcome assessment | Incomplete outcome data | Selective reporting | Other sources of bias |
|---|---|---|---|---|---|---|---|
| Guo et al. [27] | unclear | unclear | high | high | low | unclear | low |
| Fang et al. [29] | low | low | unclear | unclear | low | low | low |
| Wehmeyer et al. [34] | low | low | high | unclear | low | low | low |

According to the Newcastle-Ottawa Scale, the quality of the 4 prospective studies including at least one control group and of the 4 retrospective studies were considered as moderate quality [26, 30, 37] and the others as high quality [36, 39–42] (Table 5).

The 7 prospective studies having no control group were considered at high risk of bias.

## Discussion

To the best of our knowledge, this is the first systematic review to have investigated the effect of periodontal treatment in CKD and periodontitis patients in terms of GFR, systemic inflammation, and morbidity and mortality. Some recent systematic review and meta-analysis have demonstrated an association between renal impairment and periodontitis [11, 12, 19], and pointed out the need of RCTs investigating the effect of prevention or periodontal treatment on the severity of renal dysfunction.

Among the 14 prospective studies, only 3 were RCTs and 11 were non-randomized. The objective of the 3 randomized studies was to evaluate the effect of periodontal treatment on systemic inflammation. However, the periodontal treatment differed between each study: oral hygiene instructions and non-surgical periodontal treatment or oral hygiene instructions alone [27], oral hygiene instructions and non-surgical treatment and teeth extractions if necessary [29], or oral hygiene instructions and non-surgical treatment with adjunctive local-delivery antimicrobial therapy and teeth extraction if necessary [34]. Systemic inflammation as assessed by hs-CRP [27, 29] or IL-6 [34] was the primary endpoint. Two were restricted to ESRD patients [29, 34], whereas the third study included all patients with a GFR<60 mL/

**Table 5. Newcastle-Ottawa quality assessment scale (studies including control groups).**

| Study | Study design | Selection | | | | Comparability | Outcome | | | Total |
|---|---|---|---|---|---|---|---|---|---|---|
| | | A | B | C | D | E | F | G | H | |
| Tasdemir et al. [26] | Prospective | 1 | 1 | 1 | 0 | 0 | 1 | 1 | 1 | 6 |
| De Souza et al. [30] | Prospective | 1 | 1 | 1 | 1 | 0 | 1 | 1 | 0 | 6 |
| Vilela et al. [36] | Prospective | 1 | 1 | 1 | 0 | 1 | 1 | 1 | 1 | 7 |
| Artese et al. [37] | Prospective | 1 | 1 | 1 | 0 | 0 | 1 | 1 | 1 | 6 |
| Santos-Paul et al. [39] | Retrospective | 1 | 1 | 1 | 1 | 0 | 1 | 1 | 1 | 7 |
| Huang et al. [40] | Retrospective | 1 | 1 | 1 | 1 | 1 | 1 | 1 | 1 | 8 |
| Huang et al. [41] | Retrospective | 1 | 1 | 1 | 1 | 1 | 1 | 1 | 1 | 8 |
| Lee et al. [42] | Retrospective | 1 | 1 | 1 | 1 | 1 | 1 | 1 | 1 | 8 |

A: Representativeness of the exposed cohort; B: Selection of the non-exposed cohort; C: Ascertainment of exposure; D: Demonstration that outcome of interest was not present at start of study; E: Comparability of cohorts on the basis of the design or analysis; F: Assessment of outcome; G: Was follow-up long enough for outcomes to occur; H: Adequacy of follow up of cohorts.

min:1.73 m$^2$ [27]. Two concluded to a significant effect of the purposed treatment (even oral hygiene instructions only) after treatment compared to baseline [27, 29] or compared to the control group [29] while the third did not [34]. Despite the heterogeneity in the design of the RCTs included in this study, results supported improvement of systemic inflammation (hs-CRP) after periodontal treatment.

Furthermore, RCTs were single centre [27, 29] or single centre pilot studies [34]. Although single centre studies provide the flexibility of approach necessary for clinicians and scientists and an important source for new therapeutic ideas, they may be limited by difficulties in recruiting large numbers of participants. Another problem of such studies is the restrictive selection of studied patients that may hamper the generalization of findings.

About the 11 non-randomized prospective studies, only 4 of these included a control group [26, 30, 36, 37]. The objectives of the latter was to evaluate the effect of periodontal treatment on fatal events in ESRD patients [30], and on systemic inflammation [26, 36] or glomerular filtration rate [37] in patients with different stages of CKD undergoing conservative treatment. Despite the heterogeneity in the design of non-randomized prospective studies and therefore the difficulties to conclude on the influence of periodontal treatment on CKD, results are promising about impact of periodontal treatment on systemic inflammation and GFR.

The primary endpoint differed between studies; inflammation markers were the most frequently used. Measurement of CRP is obviously interesting owing to its association with cardiovascular disease and mortality [46], but periodontitis is an infectious disease that creates overall inflammation in active periods, including increase of CRP levels [6], and therefore improvement of inflammatory markers may not be sufficiently specific to CKD improvement. Furthermore, GFR is used to diagnose CKD and eGFR <60 ml/min/1.73 m$^2$ is associated with subsequent risk of all-cause and cardiovascular mortality, kidney failure, acute kidney injury, and CKD progression in the general population and in populations with increased risk for CVD [47]. However, this was the primary endpoint in only 2 studies. This endpoint should be measured in future studies.

In most studies, the efficacy of the purposed periodontal treatment was evaluated by measuring periodontal indices; this is a prerequisite to conclude on the influence of periodontal treatment on CKD status. However, in the included studies, the number of teeth or the capacity of chewing were not taken in account in the design of studies whereas the decrease of chewing ability will influence the diet of patients [48]. Moreover, periodontal treatment is able to prevent further tooth loss [49]. Not taking in account the number of teeth could be a bias regarding the fact that a liquid alimentation is a negative predictive risk in CKD or ESRD patients. A matching on this confounding factor can be implemented with the simple use of a questionnaire.

About the retrospectives studies, three [39–41] showed that ESRD patients receiving intensive periodontal treatment had a significantly lower risk of infections, of cardiovascular diseases and of mortality compared to those without periodontal disease. However, this could also lead to the conclusion that having a treated periodontal disease is better than no disease at all. This highlights a problem with retrospective studies for which it is practically impossible to identify and include patients who are underdiagnosed for periodontal disease.

The heterogeneity of design and protocols between studies does not allow for clear conclusions about the benefit of periodontal treatment in CKD and periodontitis patients in terms of systemic inflammation, GFR, or mortality; further methodologically robust RCTs are needed to demonstrate with a high level of evidence such potential benefit. It is of note that although there are studies that specifically investigated ESRD patients, the others did not distinguish patients on renal disease severity. Future RCTs should separate ESRD patients from those at other stages of CKD as the former are more fragile, often with haemodialysis, and are less likely

to follow PTs and regular follow-up; studies including ESRD patients should assess specifically adapted therapies, e.g. prevention or oral hygiene instructions, which can be provided in hae-modialysis centres. In addition, basic methodological aspects must be taken into account in order to minimize confounding bias, i.e. selection, attrition, performance, and detection bias. Randomization should include an adequate generation of allocation sequence associated with a clear concealment of allocation. When double-blinding cannot be obtained, a blinded assessment of the endpoints must be performed by a specific adjudication committee. The number of lost to follow-up must be minimized and handling of dropouts and missing data should be specified *a priori* in the study protocol. In order to avoid Type I Error Rate Inflation, dealing with multiplicity has to be also considered in the first versions of the protocol. Coordinated work and dialogue between nephrologist and periodontist seems obviously extremely important to set up for these future studies.

Periodontal treatment seems to improve CKD status, and results are encouraging especially for the reduction of systemic inflammation. However, further RCTs are needed to confirm these results and to specifically assess the influence of different types of periodontal treatment in CKD patients as well-documented for other systemic diseases such as diabetes [50, 51].

## Supporting information

**S1 Checklist. PRISMA 2009 checklist.**
(TIF)

## Acknowledgments

The authors thank Philip Robinson for his careful language review.

## Author Contributions

**Conceptualization:** François Gueyffier, Laurent Juillard, Brigitte Grosgogeat, Kerstin Gritsch.

**Methodology:** François Gueyffier, Delphine Maucort-Boulch, Patrice Nony.

**Supervision:** Kerstin Gritsch.

**Writing – original draft:** Théo Delbove, Kerstin Gritsch.

**Writing – review & editing:** Laurent Juillard, Emilie Kalbacher, Brigitte Grosgogeat.

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
