## [Decision Letter · Decision Letter 0]

30 Oct 2020

PONE-D-20-20992

Effect of periodontal treatment on the glomerular filtration rate, reduction of inflammatory markers and mortality in patients with chronic kidney disease: a systematic review

PLOS ONE

Dear Dr.Kerstin,

Thank you for submitting your manuscript to PLOS ONE. After careful consideration, we feel that it has merit but does not fully meet PLOS ONE’s publication criteria as it currently stands. Therefore, we invite you to submit a revised version of the manuscript that addresses the points raised during the review process.,Please address the minor revision as requested by the reviewers.,Your submission  highlights a very good topic in the field of dentistry and Nephrology .

Please submit your revised manuscript by due date , If you will need more time than this to complete your revisions, please reply to this message or contact the journal office at plosone@plos.org. Please include the following items when submitting your revised manuscript:

We look forward to receiving your revised manuscript.

Kind regards,

Bhagwan Dass, MD

Academic Editor

PLOS ONE

Journal Requirements:

2. Please ensure that the risk of bias for all the included studies (not just the RCTs) is assessed using a validated tool. For example, by using the Cochrane collaboration’s tool for randomized trials and the Newcastle-Ottawa Scale for nonrandomised studies.

3. Please ensure that references are cited in order. For example, reference 12 doesn’t appear to be cited until the discussion.

Reviewers' comments:

Reviewer's Responses to Questions

**Comments to the Author**

1. Is the manuscript technically sound, and do the data support the conclusions?

Reviewer #1: Yes

Reviewer #2: Yes

Reviewer #3: Yes

2. Has the statistical analysis been performed appropriately and rigorously? 

Reviewer #1: N/A

Reviewer #2: Yes

Reviewer #3: Yes

3. Have the authors made all data underlying the findings in their manuscript fully available?

Reviewer #1: Yes

Reviewer #2: Yes

Reviewer #3: Yes

4. Is the manuscript presented in an intelligible fashion and written in standard English?

Reviewer #1: Yes

Reviewer #2: Yes

Reviewer #3: Yes

5. Review Comments to the Author

Reviewer #1: This manuscript is a well-written review that summarize well the knowledge on CKD and periodontal treatment. It is an up to date work. I have only minor comments.

Minor:

- l40: into consideration

- l48: bacterial

- l63: through thus??

Reviewer #2: The manuscript is good and the data support the conclusions.

The right protocol was used

The statistical analysis was performed appropriately

Article will contribute to knowledge

Minor revision are required

Reviewer #3: This is very interesting article. Really well written and I could not find fault in the approach taken and written to an excellent standard. The link between chronic kidney disease and periodontal disease is one that needs to be investigated further. This article rightly draws attention to the lack of carefully designed randomised clinical trials. Something that my research group are well aware of. Conclusions were valid, this is certainly an approach that should be adopted in the future when treating renal patients that periodontal treatment is provided.

6. PLOS authors have the option to publish the peer review history of their article (what does this mean?). If published, this will include your full peer review and any attached files.

Reviewer #1: No

Reviewer #2: No

Reviewer #3: No

---

## [Author Response · Author response to Decision Letter 0]

30 Dec 2020

Response to Reviewers

Journal Requirements

The style of the manuscript has been checked and modified as requested. 

2. Please ensure that the risk of bias for all the included studies (not just the RCTs) is assessed using a validated tool. For example, by using the Cochrane collaboration’s tool for randomized trials and the Newcastle-Ottawa Scale for nonrandomised studies

The Newcastle-Ottawa Scale has been used (Table 5) to assess the quality of the other included studies (with control groups). 

Line 91 (Material and methods section), the following sentence has been added " The quality of each non-randomized study (including at least one control group) was evaluated using the Newcastle-Ottawa Scale [22]. This scale includes 3 categorical criteria with a maximum score of 9 points: the selection of the study groups; the comparability of the groups; and the ascertainment of the outcome of interest. The quality of each study was rated using the following scoring algorithms: ≥7 points was considered as “high”, 4 to 6 points was considered as “moderate”, and ≤3 point was considered as “poor” quality study."

Line 199 (Results section), the following paragraph " Among the 14 prospective studies, 7 had no control group [24,27,30-32,34,37] and only 3 were RCTs [26,28,33] and could be used for the Cochrane collaboration’s tool for assessing risk of bias in randomized trials. Two studies showed high risk of bias concerned blinding conditions (Table 4) [26,33]. " has been replaced by " Among the 14 prospective studies, 3 were RCTs [27,29,34], 4 included at least one control group [26,30,36,37] and 7 had no control group [25,28,31-33,35,38]. 

The 3 RCTs could be used for the Cochrane collaboration’s tool for assessing risk of bias in randomized trials. Two studies showed high risk of bias concerned blinding conditions (Table 4) [27,34]. 

According to the Newcastle-Ottawa Scale, the quality of the 4 prospective studies including at least one control group and of the 4 retrospective studies were considered as moderate quality [26,30,37] and the others as high quality [36, 39-42] (Table 5). 

The 7 prospective studies having no control group were considered at high risk of bias."

This modification has led to the addition of a new reference; the numbers of the other references have been modified accordingly.

3. Please ensure that references are cited in order. For example, reference 12 doesn’t appear to be cited until the discussion.

The references have been checked. Reference 12 had been erroneously removed from the introduction section. This has been corrected.

4. Please include captions for your Supporting Information files at the end of your manuscript, and update any in-text citations to match accordingly.

The caption for our Supporting Information file (S1 Checklist. Prisma 2009 checklist) has been included at the end of our manuscript. The Prisma Checklist has been modified (column "Reported on page #") in accordance with the present changes of the manuscript (revised version). 

Reviewer's Comments

Reviewer #1

This manuscript is a well-written review that summarize well the knowledge on CKD and periodontal treatment. It is an up to date work. I have only minor comments.

Minor:

- l40: into consideration

- l48: bacterial

- l63: through thus??

Thank you. As suggested, modifications have been done. 

Reviewer #2

The manuscript is good and the data support the conclusions.

The right protocol was used

The statistical analysis was performed appropriately

Article will contribute to knowledge

Minor revision are required

Reviewer #3

This is very interesting article. Really well written and I could not find fault in the approach taken and written to an excellent standard. The link between chronic kidney disease and periodontal disease is one that needs to be investigated further. This article rightly draws attention to the lack of carefully designed randomised clinical trials. Something that my research group are well aware of. Conclusions were valid, this is certainly an approach that should be adopted in the future when treating renal patients that periodontal treatment is provided.

Reviewer Comments submitted as an attachment file

Introduction

Line 52: kindly change i.e. to for example (Because there are other associations between periodontitis and systemic diseases)

Line 58: prevention of what? (Suggestion: prevention of periodontal disease

Line 63: kindly remove ‘Through thus’(Suggestion using clinical parameters)

Thank you for the suggestions. Modifications have been made accordingly.

Materials and methods

Study selection

Line 68-72: Any article where participants had other systemic diseases already associated with periodontitis like D/M, CVD, etc should have been excluded from this study

We agree that these other systemic diseases need to be taken into account. As indicated in our discussion, further methodologically robust RCTs are needed. Nevertheless, CKD patients often suffer from co-morbidities, especially ESRD patients. We have chosen, in this review, to keep all the studies in order to take into account this clinical aspect, and to help practitioners in the management of their CKD patients. 

Search Strategy

LINES 80-83: What did you do with the articles that were duplicated since there was a likelihood of that as 2 different search engines were used (PUBMED and Web of Science)?

This should be reported here as well as under study selection

The duplicates are mentioned in the results section (in the paragraph Study selection) and in the flow diagram (Fig 1) as advised by following the PRISMA guidelines.

Data collection

Line 88 /89: kindly delete risk of, if, found, involved. Kindly replace with ‘only each risk of bias……………………….in the first instance; where consensus could not be reached, a third reviewer was consulted’

The sentence has been modified as suggested.

Results

Study design

Line 126: How sure were the authors of study 25 that the result they got was not due to the periodontal effect of diabetes since a 2-way relationship had been established between periodontitis and diabetes mellitus? Shouldn’t this article have been discarded in order to avoid bias? Kindly state this as a limitation to the study

In the study 25 (study 25 has become study 26 after the addition of a reference), the experimental group (Chronic Kidney Disease + Diabetes + Periodontitis) was compared to 2 control groups, one including patients with Diabetes + Periodontitis and one including patients with Periodontitis only, taking into consideration the relationship between periodontitis and diabetes mellitus. Following the Journal requirements, the quality of this study has now been assessed using the Newcastle-Ottawa scale. 

Line 142: Assessment on renal disease and systemic inflammation: Kindly replace with ‘assessment of ………….

Periodontal assessment criteria and periodontal treatment efficacy evaluation

Line 171: kindly delete ‘for’

172: kindly replace ‘has been’ with ‘was’

267: priori………………priority

Thank you for the suggestions. Modifications have been made accordingly. 

Line 267 (line 282 in the revised version), "a priori" has been written in italics to avoid confusion

---

## [Editor Report · Decision Letter 1]

5 Jan 2021

Effect of periodontal treatment on the glomerular filtration rate, reduction of inflammatory markers and mortality in patients with chronic kidney disease: a systematic review

PONE-D-20-20992R1

Dear Dr. Gritsch,

We’re pleased to inform you that your manuscript has been judged scientifically suitable for publication and will be formally accepted for publication once it meets all outstanding technical requirements.

Kind regards,

Bhagwan Dass, MD

Academic Editor

PLOS ONE
---

## [Editor Report · Acceptance letter]

7 Jan 2021

PONE-D-20-20992R1 

Effect of periodontal treatment on the glomerular filtration rate, reduction of inflammatory markers and mortality in patients with chronic kidney disease: a systematic review 

Dear Dr. Gritsch:

I'm pleased to inform you that your manuscript has been deemed suitable for publication in PLOS ONE. Congratulations! Your manuscript is now with our production department. 

Kind regards, 

on behalf of

Dr. Bhagwan Dass 

Academic Editor

PLOS ONE